# Reciprocity in ambiguous situations: Default psychological strategies underlying ambiguity resolution in moral decision-making

Elijah Galvan[1,2]*, Alan Sanfey[1,2]

1 Behavioural Science Institute, Radboud University, Nijmegen, GE, Netherlands, 2 Donders Centre for Cognitive Neuroimaging, Radboud University, Nijmegen, GE, Netherlands

* epgalvan20@g.ucla.edu

## Abstract

When deciding whether to reciprocate trust, people are typically strongly influenced by how much trust their interaction partner has originally shown them. If a partner has placed a lot of trust in you, there is a strong motivation to reciprocate, and indeed this factor often out-weighs pro-self considerations to maximize one's own financial payout. However, one important unanswered question in this regard is what people decide to do when this prior information is ambiguous; that is, when they do not know for sure exactly how trusting their partner has been. How then do people decide to reciprocate? This study utilizes a novel version of the Trust Game to directly address this question. Here, we develop, and validate, a computational model-based approach to quantify and categorize how participants assessed the trustworthiness of an unfamiliar partner when making reciprocity decisions. We find that participants spontaneously use their prior experience about the trustingness of game partners in general to inform their reciprocity decisions, even when they had the opportunity to strategically assume that their new, unfamiliar, partners were untrusting, and hence could have justified lower reciprocation rates.

## Introduction

Interpersonal relationships rely on a reasonable expectation of reciprocity–people trust others because they have a well-founded expectation that their trust will be rewarded with reciproca-tion, rather than greedily exploited. Together, trust and reciprocity represent the foundation for cooperation, which is increasingly vital to foster in the modern age when faced with shared, existential, threats such as climate change which requires extensive, costly, and coordinated efforts to mitigate. Thus, researchers have increasingly focused on studying the how and why of trust and reciprocity. To this end, experimental studies using the Trust Game paradigm pro-posed by Berg, Dickhaut, and McCabe [1] comprise much of our working knowledge of these two behaviors and the underlying psychological motives that drive trust and reciprocity decisions.

**Data Availability Statement:** Data are available from the Donders Repository via https://doi.org/10.34973/8efg-pc38.

**Funding:** The author(s) received no specific funding for this work.

**Competing interests:** The authors have declared that no competing interests exist.

Experimental studies of reciprocity using the Berg, Dickhaut, and McCabe's Trust Game have broadly shown that people differ in their tendencies to exploit others' trust, though most people do reciprocate at the expense of maximizing their own payout. Further, the degree to which one's partner has bestowed trust strongly determines the degree to which one will forgo maximizing one's own payout at the expense of reciprocating that bestowed trust [1–3]. However, the experimental studies which have focused on reciprocity have largely overlooked a vitally important aspect of human social interactions which strongly determines the tendency to reciprocate trust–ambiguity. Ambiguity, also referred to as second-order risk, is characterized by uncertainty about the outcome for a given choice option. Real-world reciprocity decisions are riddled with such ambiguity: we rarely have clear and certain information about the extent to which trust has been bestowed or what the trustor's motives in doing so are.

For instance, if our boss has asked us to stay late at work to complete an urgent presentation for the next day, we typically do not have a guarantee that this person has actually bestowed trust in us, or if they might instead be making the same request of others so that they have multiple options to choose from. Since no previous studies have investigated how people reciprocate ambiguous trust, we do not yet have an empirical answer to this question. In fact, it is possible that people never reciprocate ambiguous trust–the mere possibility of reciprocating distrust could make people averse to reciprocating; if this is true, it would undermine the conclusions of previous laboratory studies which studied reciprocity under conditions of certainty, since these studies often show that the vast majority of people reciprocate trust. Consequently, in this study we will determine if people reciprocate trust when it is ambiguous. More specifically, since reciprocity is determined by the trust which has been bestowed, we will identify how this unknown bestowed trust is imputed in order to make reciprocation decisions.

Though there is no clear answer to this question currently available in the literature, previous work has made some intriguing suggestions. Perhaps the best indication as to how people will reciprocate ambiguous trust comes from the Offer Game [4]. The Offer Game is an amended version of the Ultimatum Game in which the endowment received by the Proposer is hidden from the Responder. The term Endowment refers to the money provided for use in the game. In the standard Ultimatum Game, the Proposer receives an endowment which is known to both parties, and then makes an ultimatum to the Responder. This ultimatum is to either accept the proposed division of assets or to reject it, in which case the Proposer and Responder receive nothing [5]. In this standard version, a Responder's decision to either accept or reject the ultimatum is typically determined by how fair–or equal—the offer is [6]. In the Offer Game, Responders therefore had first to resolve the ambiguity about the fairness of an ultimatum in order to make a decision about whether or not to accept or reject it. Unsurprisingly, Proposers made ultimatums in the ambiguous condition which Responders would have rejected in the standard Ultimatum Game, and indeed Responders were very accurate at guessing how much the Proposer was actually keeping for themselves. However, despite stating a belief that these ultimatums were unfair, Responders accepted these offers in the Offer Game at a much higher rate than they did in the standard Ultimatum Game. This raises the possibility that people might use these ambiguous situations to make strategic decisions that maximize their self-interest while preserving their self-image. Responders may be taking advantage of the ambiguity, pretending that offers are fair even when they think they are actually unfair. Since unfair offers might compel the responder to reject these offers, pretending that offers are fair can circumvent this compulsion and thereby maximize the responder's payout.

The extent to which ambiguity resolution in bargaining behavior generalizes to ambiguity resolution in reciprocity behavior is very much unclear. This is because, in contrast to the Offer Game, reciprocating bestowed trust comes at the *expense* of maximizing payout. Additionally, one's partner has no recourse against being exploited when reciprocating that

person's trust. In this sense, when bestowed trust is ambiguous–which is characteristic of real-world scenarios–the lack of certainty about what constitutes exploitation of one's partner may serve to provide justifiable cover for people to maximize their payout. Hence, to answer this question we must assess reciprocity as a function of bestowed trust, where this bestowed trust is ambiguous.

## Current study

To explore how people make reciprocation decisions when bestowed trust is ambiguous, we propose the novel Hidden Endowment Trust Game (HETG). The HETG amends the design of the standard Trust Game by making the endowment provided to the Investor hidden from the trustee, as illustrated in Fig 1. Since the Trustee does not know how much the Investor was originally given, they do not know how much trust has been bestowed upon them: the only information they have is the amount that the Investor transferred. This makes the reciprocation decision difficult, since bestowed trust is an important consideration when making reciprocation decisions. While our behavioral measure is reciprocity, the proportion of the transferred amount that the Trustee returns, the psychological outcome variable of interest is ambiguity resolution. Ambiguity resolution is defined as how Trustees assess the ambiguous bestowed trust of their partner in order to either reciprocate or exploit the Investor's trust. By making the endowment ambiguous, we can infer how people resolve the resulting ambiguity about the trust their partner has bestowed upon them.

By making the endowment ambiguous, we can learn about how people resolve the resulting ambiguity: specifically, the ambiguity about how much trust their partner has bestowed upon them. Here, we propose a specific mathematical representation for how the endowment, as a manipulated variable, affects reciprocity, as an outcome variable. A recent study by van Baar, Chang, and Sanfey [3] investigated individual differences in reciprocity motives using a modified version of the Trust Game [1]. They found that people are motivated to reciprocate trust in the context of the Trust Game from a desire to either maintain overall equity or to meet the expectations of their game partner. Crucially, irrespective of the differences in underlying reciprocity motives, the results demonstrated that trust shapes reciprocation behavior. For equity-seeking people, trust implies fairness, such that the less an Investor places trust in the Trustee, the less the Trustee should return in order to maintain equity [7–10]. However, for expectation-following people, trust is thought to convey motivations of reciprocity [11], with the prospect of violating these expectations associated with a sense of prospective guilt [12] with this aversion to guilt driving positive reciprocation behavior [13]. Consequently, reciprocation

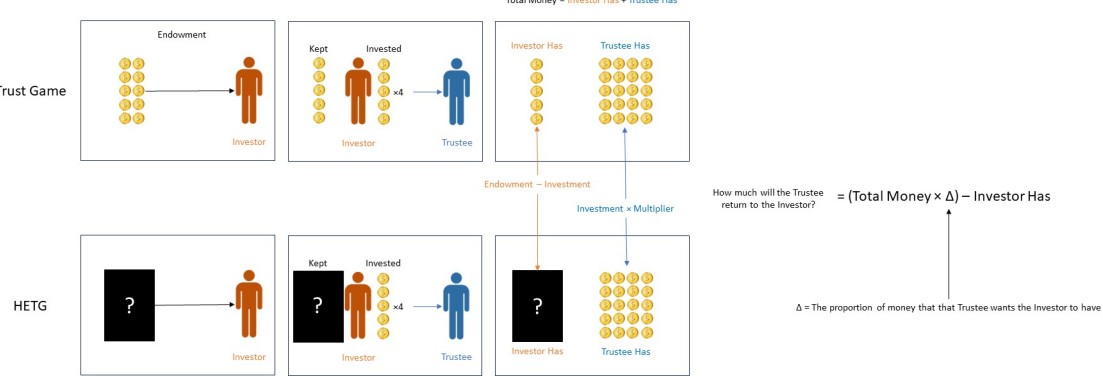

**Fig 1. Introducing ambiguity of bestowed trust in the Trust Game.**

behavior can best be modeled in the Trust Game using an Inequity-Aversion Model [14]. We have directly adapted the inequity-aversion utility model proposed by van Baar, Chang, and Sanfey [3] to be a behavioral model, which was itself adapted for the Trust Game based on earlier inequality-aversion utility models [7, 8].

Informed by previous studies which have formally modeled moral preferences in utility equations [15], the current study has formalized a behavioral model which captures inequality-aversion without a utility term, as shown in Eq 1. Here, a given Trustee's reciprocation behavior, or $R_2$, is the proportion of the money in the game (E—I is what the Investor has and $I \times M$ is what the Trustee has) that the Trustee thinks is fair for the Investor to have, minus what the Investor already has. The max argument is to enforce the game rules: namely, that the least a Trustee can return is 0: there is no 'stealing' or 'repossessing' possible. Here $\Delta$ is a free parameter that ranges from 0 to 1 and describes a specific Trustee's fairness norm–what they think a fair division of money is. Specifically, a $\Delta$ of 0 would predict that the participant never returns anything, a $\Delta$ of 0.5 would predict that the participant always returns enough so that their payout and the investor's payout are equal, and a $\Delta$ of 1 would predict that the participant always returns everything to the investor.

This approach enables us to determine how participants resolved ambiguity, since we can solve for the unknown endowment when we know their division norm. Specifically, given that M (the multiplier) is always 4 in this experiment, we can rearrange this equation to solve for E when it is ambiguous; this is shown in Eq 2. Thus, Eq 2 explains how we propose to measure ambiguity resolution in the current study. We will next focus on predicting this ambiguity resolution.

$$R_2 = \max(0, \ ((E - I + I \times M) \times \Delta) - (E - I) \tag{1}$$

$$E_{inferred} = \min((R_2 - (3 \times I \times \Delta)) \div (\Delta - 1), \ E_{max}) \tag{2}$$

Given no previous experience with, or information about, their specific game partner, we expect participants to adopt one of several ambiguity resolution strategies (ARS) to impute how much trust has been placed in them. We predict that participants will adopt one of four ARS: 1) they can give their partner the 'benefit of the doubt', and assume that the highest possible trust has been placed in them–the Highest Assumed Trust (HAT) strategy; 2) they can assume the worst, that the lowest possible amount of trust has been placed–the Lowest Assumed Trust (LAT) strategy; 3) they can assume that the particular partner plays the same strategy as the typical player they interacted with in the unambiguous version of the task–the Modal Trust (MT) strategy; 4) they could infer trust based on absolute investment amount, such that small investments imply little trust and large investments imply greater trust–the Heuristic (HR) strategy. We termed the fourth strategy as a heuristic strategy, as those that use it report employing a straightforward rule that is basically 'lower investments equal lower trust'; thus, a simple heuristic is applied to the amount that has invested to resolve ambiguity.

To assess how people actually resolve ambiguity in these situations, we will adopt a computational modeling approach. Computational models are used to capture the motives which could be used to guide decision-making, which make them a valuable tool for determining which motives guide individuals' decision-making. Thus, a computational modeling approach will enable us to directly assess if all people follow a single strategy or if people follow different strategies instead. Use of these computational approaches then allows us to test if these hypothesized motivations accurately reflect their actual decision behavior. The ARS model, shown below in Eq 4, represents the hypothesis that different people have different strategies to resolve ambiguity. Thus, the ARS model incorporates all of the potential explanations for

ambiguity resolution behavior in that it captures financially advantageous choices via the LAT strategy, captures benefit of the doubt choices via the HAT strategy, captures previous experience-informed choices via the MT strategy, and captures larger-is-more choices via the HR strategy.

## Materials and methods

### Preregistration and ethical approval

The hypotheses, sampling procedure, confirmatory analyses, and exclusion criteria for this study were registered prior to data collection on the Open Science Framework (https://osf.io/akyg7/). Preregistration is important for increasing transparency in the research process, as it clarifies distinctions between confirmatory and exploratory analyses prior to the collection of data. Thus, confirmatory results reported by preregistered studies can be trusted to not be merely exploratory results, which ensures that the False Positive Rate–or Type II error–of these results is accurately reported. Therefore, any deviation from the preregistered criteria is explicitly acknowledged.

This study was eligible for the blanket ethical approval issued to the Donders Centre for Cognitive Neuroimaging (CMO 2014/288). The blanket approval was issued to the Donders Centre for Cognitive Neuroimaging by the local ethics committee (CMO Arnhem-Nijmegen, the Netherlands). Informed consent for the study was obtained in writing.

### fMRI data

fMRI data was collected for all participants, but these results will not be reported here.

### Participants

A power analysis for the current study was conducted based on the effect size estimate of acceptance rates in ambiguous compared to unambiguous situations, from Rapoport and Sundali [4]. Thus, based on a Cohen's $d$ = 0.41, we required at least thirty-six participants to reach 80% power with a confidence level of 0.95. Accounting for a 10% exclusion rate, we aimed to collect a sample of forty participants. Left-handed participants were excluded, as were participants with any non-removable metal objects implanted in their body. In the preregistration, we indicated that we would exclude participants who ever studied psychology, economics, or neuroscience, though prior to data collection we decided to only exclude students of the local Behavioral Science Research Master.

The sample for the present study was collected based on convenience and consisted primarily of students from the Nijmegen area. A total of forty participants participated in the study–data collection began on 16 February 2022 and ended on 5 May 2022. Of these, four were excluded for expressing disbelief in the task and two were excluded as they did not complete the task. Participants who expressed disbelief in the task were excluded since these participants' decisions were unlikely to reflect their actual social preferences, undermining the validity of the conclusions which can be drawn from their data. Therefore, our final sample for analysis consisted of thirty-four participants, of whom twenty were female (59%). Twenty-eight were aged 18–30 (82%), four were aged 31–45 (12%), and two were aged 46–60 (6%).

### Procedure

The experiment consisted of a single session. Participants were brought into the MRI lab control room and asked to complete a screening form and an informed consent form. Informed consent was obtained in writing. After subjects consented to participate in the study, passed

the screening, had removed all metal from their body, and used the restroom, participants were brought into the scanning room by the experimenter. The participant was then instructed that he/she would play the 'Investment Game' exclusively as Player B (the trustee) with other Players A (the investor) who had previously completed the experiment. Our rationale for this choice was that by framing the decisions as 'investments' we might avoid the possibility that reciprocity was a consequence of a social desirability effect. Participants were given clear instructions for the Trust Game, and then played a handful of practice trials to ensure that they understood how payouts were determined for both players based on the 1) Investment amount and 2) the Returned amount. Participants were informed that one randomly selected trial would determine their additional payment, as well as their partner's: participants were informed that they would be paid at least €20 for their participation and could earn up an additional €16 based on their decision and the decision made by Player A, though no specific information was given as to how this bonus would be calculated. Bonuses were calculated based on the reciprocation decision on one randomly selected trial based on the following equation: ((Participant's Payout) ÷ (Total Money in Game)) × €16. The Investor's choice behavior was programmed by the experimenters for the purpose of the present study. After receiving the instructions, participants were instructed to place earplugs into their ear and were then placed into the MRI scanner with the response boxes, as well as a panic button that they could press to end the experiment immediately. Participants played 40 rounds of the Trust Game after which they had a short break. Then, participants were given instructions presented via the stimulus program about the HETG. Following this, they then played 80 trials of the HETG.

## Task

Participants played a total of 120 trials of the Trust Game: 40 in the unambiguous condition (i.e. the standard Trust Game) and 80 in the ambiguous condition (i.e. the HETG). The task design is shown below in Fig 2. First, the participant saw the Partner Screen which contained a blurred image of their partner's face. They then saw the Investment Screen which differed between conditions: in the unambiguous condition, the Investment was shown as a proportion of the Endowment and the amount that the participant receives was shown below as the Investment multiplied by 4. Here, the trials were designed to provide participants with a distribution of Investor trust behavior (namely that most Investors transfer half of the Endowment) and also distribution of Endowment probabilities (namely that the most common Endowment in the Trust Game is 10). See S1 Table for the full condition list in the unambiguous condition. The task was programmed in PsychoPy 3 [16].

In the ambiguous condition, Trustees are told that the endowment could be one of three possible values, which are selected based on what a possible endowment could have been, given what the Investor invested. These endowments were selected such that Bestowed Trust (Investment/Endowment) was Low (0.05–0.25), Medium (0.35–0.50), or High (0.70–1.00). Here, 20 trials showed the Investment equal to 2 and the possible Endowments equal to 2, 5, and 10; 20 trials showed the Investment ranging from 4 to 5 and possible Endowments equal to 5, 10, and 20; and 40 trials showed the Investment ranging from 7 to 10 with the possible Endowments equal to 10, 20, or 50. Participants then saw the response screen, on which they adjusted a slider to determine how much to return to their partner. The distribution of trials across both conditions was identical for all participants and was randomized; however, participants always played the unambiguous condition before the ambiguous condition in order to ensure they had previous experience upon which to base their decision-making. We programmed Investor's choices so that we would have a complete set of trials which was diverse

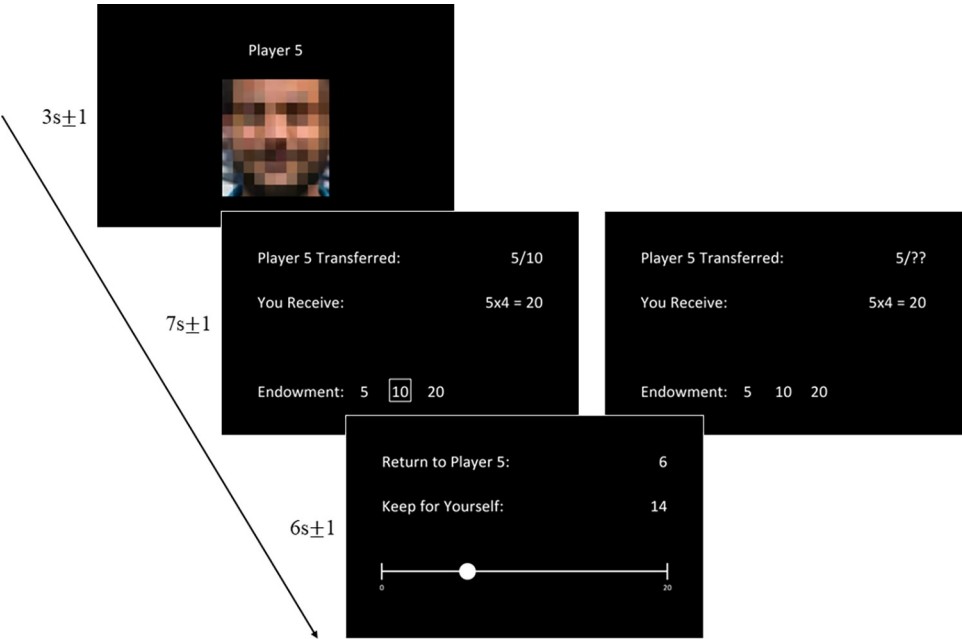

**Fig 2. Task design.** Subjects play 40 trials of the unambiguous condition (left) before playing 80 trials the ambiguous condition (right). In both conditions, subjects first see a blurred image of their partner. Then, the Investment is shown: either the Endowment is known in the Unambiguous Condition (left) or unknown in the Ambiguous condition (right) with three possible values. Three Endowments are shown in the Unambiguous condition in order to best control the visual stimuli across conditions. Subjects respond on the final screen by adjusting a slider to determine how much to return in both conditions.

enough to ensure that we could recover our free parameters. Conditions were played sequentially to remove the possibility of learning effects: if conditions were intermingled, participants would learn more about how trusting participants are on a trial-by-trial basis. Thus, this learned information has the potential to influence ambiguity resolution on a trial-by-trial basis–to avoid these contamination effects, we opted to have the Trust Game and HETG played sequentially.

## Measures

**Reciprocity.** Reciprocity was measured behaviorally in the context of the Trust Game or HETG as the proportion of the money received that participants then returned to the investor.

**Ambiguity resolution.** Ambiguity resolution was assessed using a computational modeling approach. First, reciprocity was modeled in the standard Trust Game in order to determine how participants typically reciprocate when the information is unambiguous. Then, this equation was rearranged to solve for an ambiguous endowment in the HETG (Eq 2 above).

**Manipulation check.** After completing the experiment, participants completed a survey in which they responded to a single item asking if they believed everything the experimenter told them. If participants replied "no", a follow-up question asked them to indicate what the experimenter told them that they did not believe. Participants who indicated that they did not believe that their actions would affect the payout of their partners were excluded from the analysis.

**Model fit.** Model performance was assessed via the Akaike Information Criterion (AIC), as per Eq 3 below. The AIC is considered superior to the BIC when the true data generation process is not believed to be fully represented in the model set which is the case here [17]. The

AIC weights model fit against model complexity. Model fit is represented as the sum of squared differences between predicted and observed behavior which is SSE in the model. Model complexity is captured as the number of free parameters, represented in the equation as $k$. The number of trials is given as $n$.

$$\text{AIC} = n \times \ln(\text{SSE} \div n) + k \times 2 \qquad (3)$$

## Model specification

**Mixed effects modeling.** In order to identify group-level trends in reciprocation behavior as a function of bestowed trust while controlling for individual differences, we adopted a mixed effects modeling approach. We used R [18] to conduct all analyses. We used the *lme4* package [19] to fit a linear mixed effects model: in this model, trust and the investment between trust and investment, were taken as fixed effects used to predict reciprocity ratio (i.e. Amount Returned/Pot). Here, trust was categorized as low (Investment/Endowment < 0.33), medium (0.33 < Investment/Endowment < .66), high (Investment/Endowment > 0.66), or ambiguous (Endowment not known) and we opted for a sum-to-zero contrast scheme. The sum-to-zero contrast scheme was chosen since it makes the condition effects uncorrelated with each other. Investment values were centered. Including intercept, trust, and trust-investment interaction terms as random effects resulted in a singularity warning, so the model was re-estimated without the trust-investment random effect which resolved the warning. In this case, it seems that the fixed-and-random effects for trust-investment were explaining the same variance: there were no individual differences in the trust-investment slope. We conducted a Type 3 ANOVA for this model: here, we computed p-values for this model using the Kenward-Rogers method via the mixed function in the *afex* package [20]. We conducted post hoc analyses using the *emmeans* package [21]: we compared means between trust levels using the emmeans function wrapped with the pairs function, and we compared investment slopes between trust levels using the emtrends function. The marginal $R^2$ and conditional $R^2$ values were calculated for this model using the MuMIn package [22].

**ARS model.** We estimated parameters for each subject using the Negative-Log Likelihood method. We simulated a 10201 (101 x 101) point parameter space for the ARS model where Tau and Psi ranged from 0 to 1. We computed both predicted-and-observed preference for each pair of coordinates in the parameter space using Eq 4, likelihood using the dnorm function from the *stats* package [18] in R. Deviance was then calculated as negative two times log likelihood: the free parameters selected per participant were the ones which minimized deviance in terms of expected versus observed preference.

$$p_2(\text{E}) = (1 - \text{abs}(\text{I/E} - \tau)) \times (1 - \psi) + (1 - \text{abs}(\text{I/E} - \text{I}/10)) \times (\psi) \qquad (4)$$

The ARS model captures individual differences via its free parameters. It captures participants who make decisions using the heuristic strategy with its $\psi$ parameter such that high $\psi$ values make it important that the trust implied by a given endowment agrees with the trust implied by the investment alone (i.e. $1 - \text{abs}(\text{I/E}) - (\text{I}/10)$). Inversely, low $\psi$ values make it important that the trust implied by a given endowment agrees with the participant's default assumed trustingness–or $\tau$. Therefore, when $\psi$ is high, the model predicts that participants will follow the HR strategy. When $\psi$ is low on the other hand, $\tau$ represents the strategy: values of $\tau$ close to 0 predict the LAT strategy, values of $\tau$ close to 1 predict the HAT strategy, and values of $\tau$ close to 0.5 predict the MT strategy.

## Results and discussion

### Computational modeling

**Reciprocity model.** We estimated parameters for each subject using the Negative-Log Likelihood method. The behavioral Inequity-Aversion model, given in Eq 1, explained 69.88% of the variance in participants' reciprocation behavior in the unambiguous version of the task. The model predictions were plotted against observed values, revealing that, although the model provided a reasonable fit to all data points across all ranges of predicted values, for some participants this fit was substantially worse than for others, particularly at lower predicted reciprocation values (S1 Fig). Similar to previous studies, these variations in model fit reflect the notion that reciprocation behavior in the Trust Game is difficult to model for all participants in the same way. Specifically, these variations arise since divisions can be influenced by the multiplied investment, particularly when a partner is untrusting. Thus, for a small minority of participants, the model overpredicts reciprocation amounts. The model predictions for the ambiguous trials, using the inferred endowment, explained 74.06% of the variance in reciprocation decisions, which provides some confidence regarding the validity of the inference method. Given that this does not lead to a dramatic difference in variance explained in the behavioral reciprocity model, we can be confident that the reverse inference technique is not, itself, accounting for a concerning amount of variance which is not captured in the reciprocity model.

Initially, we hypothesized that participants would learn about the likelihood of a given endowment but, based on self-report measures, it appeared participants who followed this strategy were resolving ambiguity using the heuristic strategy: therefore, we altered the computational model from the pre-registered model to more accurately reflect the psychological processes underlying the adoption of the Modal Endowment/Heuristic Strategy. This alteration did not affect the model predictions in a discernable way.

**Ambiguity resolution.** We tested the ARS Model against each individual strategy model using uncorrected, one-tailed paired sample t-tests. We excluded one participant whose behavior was perfectly predicted by both the MT and ARS Models. Our analyses showed that, in comparison to the best-fitting single-strategy model which was the MT model (Mean AIC = -276.100, SD AIC = 51.622), the ARS Model (Mean AIC = -284.662, SD AIC = 52.599) best captured ambiguity resolution behavior across the entire sample–however, this difference was not significant (Paired $t(32) = -1.352$, $p = 0.093$) and the effect size of this difference was small (Cohen's $d = 0.239$). The ARS Model was also found to substantially outperform the HAT Model (Mean AIC = -231.488, SD AIC = 68.483; Paired $t(32) = -6.541$, $p < 0.001$), LAT model (Mean AIC = -100.957, SD AIC = 30.956; Paired $t(32) = -13.956$, $p < 0.001$), and the HR model (Mean AIC = -147.808, SD AIC = 23.614; Paired $t(32) = -16.087$, $p < 0.001$). Fig 3 shows the single-strategy models plotted with 95% CIs relative to the ARS Model. Our justification for not correcting for multiple tests was that one single-strategy model would fit all participants better than the other three single-strategy models so, ultimately, we would be testing the ARS model against only the best-fitting single-strategy model.

### Mixed effects modeling

The mixed effects model was found to explain a substantial portion of the variance in reciprocity decisions. The variance explained by the model as a whole (Marginal $R^2$) was 72.478% while the variance explained by the fixed effects alone (Conditional $R^2$) was 51.395%. We found that reciprocity differed significantly as a function of trust ($F (3, 32.84) = 27.75$, $p < 0.001$) and this effect differed as a function of investment ($F (4, 4056) = 11.79$, $p < 0.001$).

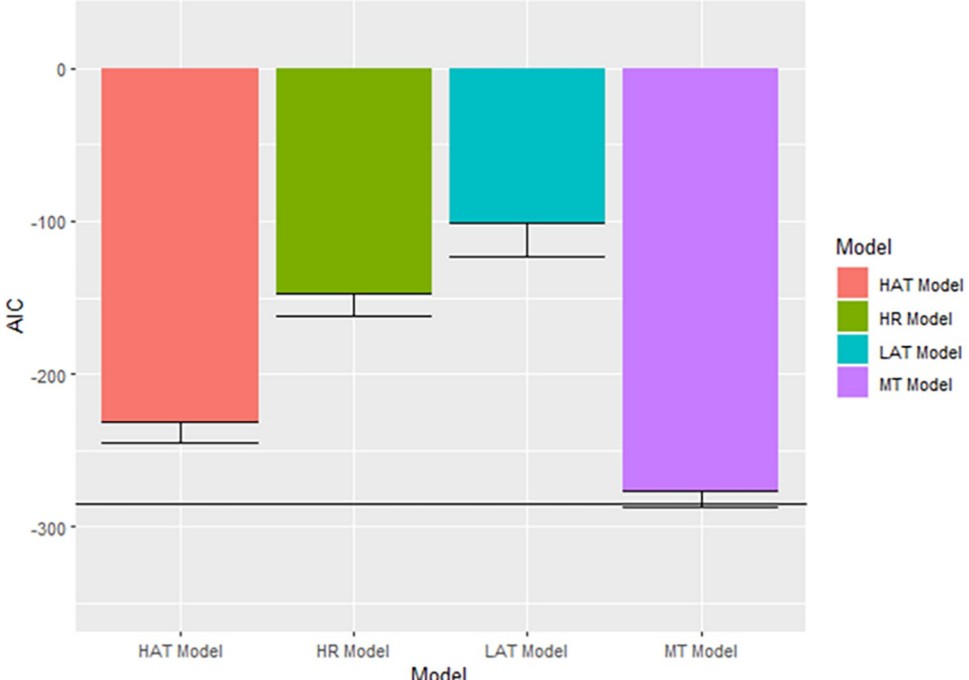

**Fig 3. AIC for the ARS model versus its constituent strategies.** More negative AIC values reflect better model fit. The horizontal line refers to the mean AIC of the ARS Model. Error bars reflect one-tailed, uncorrected 95% Confidence Intervals.

Comparing group means, we found that all pairwise contrasts were significant apart from the middle-ambiguous contrast–a finding which is unsurprising based on the results of the computational modeling analysis. These results are shown below in Table 1 and visualized in Fig 4. We further found that the effect of Investment was only significantly different between the ambiguous and high trust conditions (*Estimate* = 0.008, *SE* = 0.003, *t* (4056) = 2.837, *p* = 0.027) and the medium trust and high trust conditions (*Estimate* = 0.009, *SE* = 0.003, *t* (4056) = 3.058, *p* = 0.012).

## Conclusions

The main hypothesis of the current study was that people would differ in the strategy that they used to resolve ambiguity in order to reciprocate trust. Our data do not support this hypothesis; instead, based on our computational modeling results we concluded that participants generally resolved ambiguity in the same way. Namely, participants spontaneously used their previous experience to guide their ambiguity-resolution. This accuracy orientation illustrates

**Table 1. Pairwise mean differences for trust conditions.**

| Contrast | Estimate | SE | p-values |
|---|---|---|---|
| Low-Medium | -0.124 | 0.018 | <0.001 |
| Low-High | -0.231 | 0.025 | <0.001 |
| Medium-High | -0.107 | 0.013 | <0.001 |
| Low-Ambiguous | -0.102 | 0.018 | <0.001 |
| Medium-Ambiguous | 0.022 | 0.010 | 0.166 |
| High-Ambiguous | 0.129 | 0.017 | <0.001 |

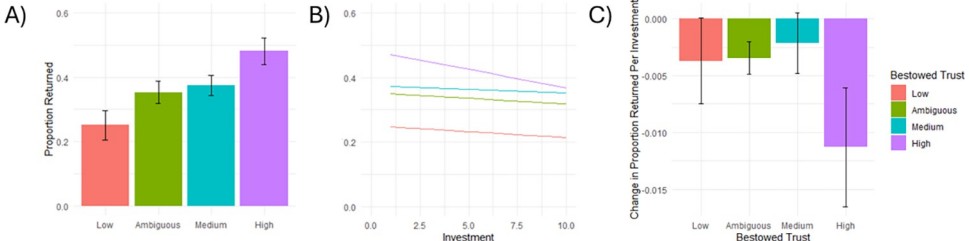

**Fig 4. Effect of bestowed trust on reciprocation decisions.** Emmeans for each Bestowed Trust Condition are plotted in A which show that, predictably, Bestowed Trust positively predicted Reciprocation Behavior. In B, we can see that greater Investments–resulting in larger pots–lead to less Reciprocity. In C, we can see that the slope of Investment was significantly different between the High and Medium Bestowed Trust Conditions. Across all graphs, the trends which were true of the Medium Bestowed Trust condition was also true of the Ambiguous Bestowed Trust condition and that the Medium and Ambiguous Bestowed Trust Conditions were not significantly different from each other on any measure.

an aversion to potentially exploiting this bestowed trust to thereby maximize their financial payout. This finding is contrary to the self-interest maximizing behavior observed in the Offer Game [4]. In the Offer Game, people were highly accurate in their estimates of the ambiguous endowment, though they did not use this information in their decisions to respond to the ultimatums. We propose that these differences are due to the nature of the respective tasks: according to utility theory, rejected offers are offers under which the utility experienced from the payout under the ultimatum, when combined with the disutility experienced from accepting the unfair offer, is less than the utility experienced from rejecting the unfair offer. Therefore, the results from the Offer Game must be attributed one or more of the following motives: 1) ambiguity resolution is biased by a desire to maximize payout, 2) the disutility experienced from accepting unfair ultimatums decreases under ambiguity, or 3) experienced utility from rejecting unfair ultimatums decreases under ambiguity. Both our computational modeling results and linear mixed effects modeling results strongly indicate that these motives are not at play when considering ambiguous trust, rather than ambiguous bargaining behavior. Our working interpretation of these results is that ambiguity resolution in the Offer Game is not different from ambiguity resolution in the HETG. Instead, the uncertainty in the Offer Game decreases the utility associated with rejecting potentially unfair offers, while the uncertainty in the HETG does not significantly impact the prospective utility of reciprocating bestowed trust. This explanation explains how behavior in the HETG reflects an accuracy orientation in resolving ambiguity while behavior in the Offer Game reflects a payout maximization motive.

In the context of the larger body of work on social decision-making under ambiguity, our results are somewhat anomalous: many researchers find that decision-making under ambiguity is less prosocial. Therefore, we must consider what specifically about the HETG distinguishes it from the tasks used in these previous studies. For example, in both the Public Goods Game (PGG) and Prisoner's Dilemma (PD), wherein participants must make the decision to cooperate or defect without knowing their partner's decision or past behavior, it has been consistently reported that participants cooperate less often than they defect [23–25]. In a version of the PGG wherein the group must cooperate to reach a threshold of contributions to the community pot or otherwise lose all their money, participants cooperate more; however, when ambiguity is introduced about what this threshold is, participants cooperate much less [26]. In this regard, it is possible that the specific ambiguity which is present in the HETG has a diverging effect because behavioral distrust is incentive-incompatible. In other words, participants in the HETG know that, regardless of any beliefs that they might have about how trusting investors are in general, their partners will invest more if they trust the participant more. Indeed,

previous research on the PD suggests that perceptions of morality are insensitive to outcomes [27]. In his light the results from the HETG may therefore be attributed to a general aversion to behaving in what may be regarded as an immoral way simply because the context of the HETG does not offer sufficient cover to morally justify behaving selfishly. Concordantly, Tho Pesch and Dana [28] find that charitable giving decreases when participants must make a choice about which charity to donate to before making the giving decision, rather than just making only the giving decision. The obfuscation of the charitable choice by simply providing attributes for comparison has been referred to as attributional responsibility [29]. In regard to the HETG, the lack of attributional ambiguity–or the inability to, perhaps, attribute ostensibly selfish behavior to anything other than a response to distrust, which was unmerited based on their previous experience, may be responsible for this atypical pattern of behavior.

The finding that ambiguity-resolution follows an accuracy orientation is in line with many studies which suggest that social behavior–such as reciprocity–is motivated by a desire to act in a morally correct manner. While our study design was not suited to investigate the intentions underlying prosocial behavior–reciprocity in this instance–the notion that such behavior arises from a preference for behaving in a morally correct manner rather than a desire to achieve a certain outcome–namely, the moral preferences hypothesis–is a powerful lens through which to view our findings [30]. This is because in scenarios where distribution outcomes are ambiguous–notably the case in both our task and reciprocity in the real-world–prosocial behavior must then be guided by principles used to distinguish the morally right from the morally wrong. Through such a perspective, adopting an accuracy orientation for ambiguity resolution can be regarded as evidence for trustees to reciprocate based on interpretations of this ambiguity made in good faith.

When considering how the results we reported came to be, it should be noted that participants were not informed about the rules of the HETG prior to playing the unambiguous, standard Trust Game. It is possible that some, or all, of these participants used the fact that three endowments were presented in the standard Trust Game–which we included to control the visual stimuli across conditions–to deduce that they would be playing a version of the game in which this information would be uncertain. Since we did not have a self-report item to either confirm or refute this interpretation, we cannot conclude with certainty that this is not the case. Regardless of whether this information was recalled spontaneously or if participants were intentionally tracking trustingness while playing the unambiguous trials, participants behaviorally demonstrated an orientation to accurately infer the ambiguous trust their partner placed in them and used the information they previously passively accrued to inform this inference. Thus, since these results have shown that positive reciprocity is not simply a consequence of disambiguating bestowed trust in the laboratory setting, our findings bolster the ecological validity of the standard Trust Game as a means by which to study positive reciprocity.

## Reflection on null results

Our analyses did not support the preregistered hypothesis that there are significant individual differences in default ambiguity resolution strategies, since the ARS model's AIC was not significantly lower than the MT model's, though this difference was somewhat close to statistical significance. Null results notwithstanding, the ARS model performed better than MT model when accounting for the decrease in parsimony, which indicates that a few participants did indeed use different strategies. At the individual level, we found that a majority of people (71%) used the MT strategy to resolve ambiguity while the next most prevalent strategy was the LAT strategy (23%). Therefore, it seems that a significant minority of our sample did indeed take advantage of ambiguity in order to make strategic decisions that maximize their

self-interest while preserving their self-image. The adoption of the LAT strategy may evince some kind of motivated reasoning wherein participants adopt it because they are motivated to reach the conclusion that they should keep as much as possible [31]. In this regard, given that participants prior experience was so strongly indicative that the average investor bestowed a 'medium' amount of trust, participants who we identified as being MT strategists may have adopted an accuracy-orientation because this motivated reasoning process–which would result in the adoption of the LAT strategy–could not override the more accurate one.

Given that the data show that there are individual differences in ambiguity-resolution behavior, but these differences are not significant, we believe our null findings are, in part, due to the underpowering of our study. Our power analysis, which was based off an effect size of 0.41 from [4], indicated that 36 participants would be sufficient to reach 80% power. Importantly, the results from this study indicate that ambiguity resolution in the Offer Game is different from ambiguity resolution in the HETG and the achieved power computed post hoc from this study is just 36%. Thus, we believe that this study was underpowered for revealing individual differences in ambiguity resolution when making reciprocation decisions.

## Limitations

The most central limitation of the current study is how little ecological validity the experimental design used in the current study has. In order to validate our reverse inference technique, we needed participants to complete dozens of trials with new partners who they knew nothing about–with the exception of a blurred picture and an investment amount, there was no information which participants could base their decision off of. Instead, they would have to default to a strategy to resolve ambiguity: a strategy which we could very clearly identify in the data. However, this experimental control comes at the cost of generalizability: in real-world settings, people always know at least something about the person they are trying to infer the expectations, beliefs, or intentions of. Whether this is something as irrelevant as facial features or as relevant as reputation or past experience, these factors often influence social decision-making [32–35]. The current study is very limited in the sense that in an anonymized one-shot design it controls and eliminates the effect of such factors on reciprocation behavior and, by extension, ambiguity-resolution. Therefore, it is very much unclear how these ambiguity resolution strategies actually impact reciprocation decisions made under ambiguity, since it removes these other variables from consideration.

A further limitation of the current study is the population which the sample was drawn from. Since our sample was comprised nearly exclusively of university students, the conclusions we can draw from our data are inherently constrained by the fact that we have sampled a portion of the adult population which is rather young, often of a relatively high SES, and share a similar educational background. For instance, respondents' propensity to use an accuracy-oriented approach to resolve ambiguity may not be characteristic of a broader population who may be less idealistic and who, being generally less privileged, may be more self-interested. In a similar regard, the current study was conducted on a WEIRD population: WEIRD populations–which are Westernized, Educated, Industrialized, Rich, and Democratic–comprise a disproportionate majority of research on human psychology [36, 37]. The current study falls into this pattern: convenience samples are often used in the first stages of research, but further research into human decision-making under ambiguity must be done with a more diverse sample than we utilize here.

Another shortcoming is the sample size–as previously mentioned in our reflection on the null results of this study, our study was likely underpowered since the best effect size available was much larger than the actual effect size of the current study. This small sample size may

therefore be responsible for our inability to detect meaningful heterogeneity in our sample–heterogeneity which might actually be present. Consequently, without adequate power our results do not offer conclusive evidence that these individual differences exist or if, instead, participants actually all follow the same strategy to resolve ambiguity.

A final limitation is in the variations observed in model fit. This is a common finding when modeling reciprocity in the context of the Trust Game since division norms can be applied to all of the money in the game–as in our model–but, alternatively, to the multiplied investment alone or a combination of the two. Thus, this heterogeneity is not accounted for in the model, which explains why a minority of participants' data was more poorly explained at reciprocation values further from the mean. Pragmatism notwithstanding, this calls into question the ability to make reliable reverse inferences for this minority of participants whose data was overpredicted at low trust values and underpredicted at high trust values.

## Strengths of this study

**Capturing individual differences.** A substantial strength of this study was its use of computational modeling to capture individual differences. The conventional approach to modeling behavior is descriptive, mean-based linear modeling: such an approach captures sample-level trends and accounts for individual differences using a priori predictors. In other words, such an approach only detects individual differences which are explained by prespecified variables. The current approach differs: instead of trying to describe the data generation process, the current approach predicts it a priori. Thus, this data generation process incorporates the individual differences we expect to see without having to specify *why* some people behave differently than others. Our approach to classifying strategies was based on the grouping of a model predictions of an equation which was preregistered: therefore, our approach did not require us to speculate about what would predict the adoption of these strategies. Thus, our approach enabled us to get a clear and convincing account of how individuals actually resolved ambiguity in order to make reciprocation decisions.

**Reverse inference technique.** Another substantial strength of the current study was that it utilized a novel reverse inference technique to assess how people resolved ambiguity in the context of the HETG. The success of this technique was supported by several findings. First, we found that the strategies identified using the ARS model were consistent with those described by participants via self-report. Second, the behavioral trends observed in the medium (non-ambiguous) trust condition were not significantly different from those in the ambiguous condition. Since the model comparison indicated that accounting for strategies other than the Modal Trust strategy was not justified by the loss of parsimony, this further supports the conclusion of the model comparison, and also suggests that the computations underlying reciprocity do not necessarily change as a function of ambiguity. Third, the behavioral trends between the medium and the low/high bestowed trust conditions was also true of the ambiguous bestowed trust condition. This further supports the notion that the computations underlying reciprocity do not necessarily change as a function of ambiguity–rather, people simply impute the ambiguous value in the otherwise unchanged computation underlying reciprocity. Fourth, when bestowed trust was unambiguous, reciprocation rates significantly differed as a function of bestowed trust. This validates a fundamental assumption of the current study–that by making the bestowed trust ambiguous, we were impacting a crucial decision variable in the computation underlying reciprocity. Finally, the variance explained by the computational model of reciprocity was not substantially different between the unambiguous and ambiguous conditions. This eliminates the alternative explanation that the model is simply fitting *more* noise. Thus, we are confident that valid and reliable conclusions can be drawn using the reverse inference technique.

Ambiguity has been largely ignored in the study of reciprocity despite the fact that is ubiquitous in social interactions. Researchers often must choose between utilizing field studies–where there is little control over the variables determining behavior–or laboratory studies–where the control over the variables determining behavior is perhaps too rigid to be confident that one's findings generalize to real-world behavior. We believe that bringing theoretical, a priori, computational approaches into the study of social behavior will enable researchers to answer broader, more compelling questions about the psychological mechanisms underlying social behavior than was previously possible in the laboratory setting using descriptive, post-hoc, linear modeling approaches. Here, our use of a novel approach to computationally drawing reverse inferences serves as an example of how one can answer such questions. More specifically, our approach to modeling and inferring about ambiguity resolution will enable future research to study more about how people resolve ambiguity about others' intentions in order to make decisions that concern themselves and that other person.

## Future directions

Overall, the current study has many limitations which should be taken into account when drawing conclusions about the results. A clear future direction of the current study would be a replication with a large sample–a large enough sample to indicate if individual differences do exist in ambiguity resolution strategy adoption or if there are no meaningful individual differences after all. Beyond this, the novelty of the current study's design and analysis approach open up new opportunities for future research. In this light, we will focus on the potential to increase the ecological validity of research on reciprocity. For example, using the reverse inference technique to measure ambiguity resolution, future research can study how ambiguity resolution changes with repeated interactions. Similarly, future studies could investigate how factors such as facial features or word of mouth influences how people resolve ambiguity. In both scenarios, this reverse inference technique can be applied to study social behaviors beyond merely reciprocity. However, as it pertains to reciprocity specifically, such research would clarify how influential these default ambiguity resolution strategies actually are for making reciprocation decisions in real world scenarios: do they serve as an anchor-and-adjust heuristic or do they instead only emerge when no other information is available.

## Supporting information

**S1 File. Video abstract.** A brief video explaining our research question, experimental design, and main finding.
(MP4)

**S1 Table. Distribution of unambiguous trials.**
(XLSX)

**S1 Fig. QQ plot behavioral reciprocity model.** The behavioral reciprocity model provided a good fit to the data ($R^2 = 0.722$), though model performance at higher and lower predicted values was worse than expected.
(TIF)

## Author Contributions

**Conceptualization:** Elijah Galvan, Alan Sanfey.

**Data curation:** Elijah Galvan.

**Formal analysis:** Elijah Galvan.

**Investigation:** Elijah Galvan.

**Methodology:** Elijah Galvan.

**Resources:** Alan Sanfey.

**Supervision:** Alan Sanfey.

**Validation:** Elijah Galvan.

**Visualization:** Elijah Galvan.

**Writing – original draft:** Elijah Galvan.

**Writing – review & editing:** Elijah Galvan, Alan Sanfey.

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
