## [Decision Letter · Decision Letter 0]

23 Oct 2023

PONE-D-23-24513Reciprocity in Ambiguous Situations: Default Psychological Strategies Underlying Ambiguity-Resolution in Moral Decision-MakingPLOS ONE

Dear Dr. Galvan,

Thank you for submitting your manuscript to PLOS ONE. After careful consideration, we feel that it has merit but does not fully meet PLOS ONE’s publication criteria as it currently stands. Therefore, we invite you to submit a revised version of the manuscript that addresses the points raised during the review process. Please address the comments made by both reviewers in the revised manuscript and clearly indicate the changes that were made while addressing them. 

We look forward to receiving your revised manuscript.

Kind regards,

Inon Zuckerman

Academic Editor

PLOS ONE

Journal Requirements:

2. Thank you for submitting the above manuscript to PLOS ONE. During our internal evaluation of the manuscript, we found significant text overlap between your submission and previous work in the [introduction, conclusion, etc.].

Please revise the manuscript to rephrase the duplicated text, cite your sources, and provide details as to how the current manuscript advances on previous work. Please note that further consideration is dependent on the submission of a manuscript that addresses these concerns about the overlap in text with published work.

[If the overlap is with the authors’ own works: Moreover, upon submission, authors must confirm that the manuscript, or any related manuscript, is not currently under consideration or accepted elsewhere. If related work has been submitted to PLOS ONE or elsewhere, authors must include a copy with the submitted article. Reviewers will be asked to comment on the overlap between related submissions (http://journals.plos.org/plosone/s/submission-guidelines#loc-related-manuscripts).]

We will carefully review your manuscript upon resubmission and further consideration of the manuscript is dependent on the text overlap being addressed in full. Please ensure that your revision is thorough as failure to address the concerns to our satisfaction may result in your submission not being considered further.

Reviewers' comments:

Reviewer's Responses to Questions

**Comments to the Author**

1. Is the manuscript technically sound, and do the data support the conclusions?

Reviewer #1: Partly

Reviewer #2: Yes

2. Has the statistical analysis been performed appropriately and rigorously? 

Reviewer #1: I Don't Know

Reviewer #2: Yes

3. Have the authors made all data underlying the findings in their manuscript fully available?

Reviewer #1: No

Reviewer #2: Yes

4. Is the manuscript presented in an intelligible fashion and written in standard English?

Reviewer #1: Yes

Reviewer #2: Yes

5. Review Comments to the Author

Reviewer #1: Review of Manuscript

The paper presents a novel exploration of ambiguity in reciprocity decisions, employing a computational modeling approach. While the study provides valuable insights into how individuals handle uncertainty, several areas can be refined. The entire paper, especially the Methods section, would benefit from more explicit explanations. It's important to delve deeper into participant selection criteria and elaborate on experimental design choices for better reader understanding. In the Results section, enhancing the systematic presentation of findings and aligning them with the study's objectives would enhance clarity. The Discussion section could be improved through more extensive comparisons with existing literature and stronger connections to the study's objectives and hypotheses. Despite these areas for improvement, this research lays a promising foundation for further investigating ambiguity in social decision-making. It underscores the importance of future studies considering individual differences in decision strategies. Below, you will find my comments for each section of the manuscript.

Introduction

1. Explicitly state the literature gap the study addresses. Identify unanswered questions regarding ambiguity in trust to set a focused research context. What unanswered questions or uncertainties about ambiguity in trust exist in the current research landscape? Highlight the study's key contributions to establish clear expectations for readers.

2. Consider relocating equations, like [Disp-formula pone.0300886.e003], to the Methods section. This streamlines the introduction, making it more accessible. Provide concise explanations of specialized terminology and equations for broader readership. Add transitional sentences to link the introduction to subsequent sections.

3. Provide a concise explanation of the ARS model and its relevance. Establish a clear link between the Offer Game discussion and the ARS model introduction.

4. Elaborate on the "veil of ambiguity" concept's relevance and implications for the research. Explain its connection to the Trust Game and study objectives.

5. Simplify complex sentences and explanations for better readability and understanding.

6. Consider using diagrams or graphs to illustrate how ambiguity is introduced in the Hidden Endowment Trust Game.

Methods

7. Briefly explain the importance of preregistration for research transparency and bias reduction.

8. Clarify the exclusion of left-handed participants and those with non-removable metal objects. Address whether these were the sole exclusion criteria for fMRI participants. Justify excluding local Behavioral Science Research Master students and discuss its impact on generalizability.

9. Provide context for the sequential Trust Game and HETG. Explain its influence on the experimental design.

10. Describe participant training or instructions for the Trust Game and HETG to shed light on decision-making processes.

11. Provide a detailed explanation of the payment structure, outlining the specific criteria and mechanisms through which participants could earn the additional €16 based on their decisions and Player A's choices.

12. Elaborate on the reasoning behind programming the Investor's choices. This additional information will enhance readers' comprehension of the programming's purpose and its importance in the study.

13. Explain how the measures address research questions and why computational modeling suits ambiguity-resolution.

14. Clarify why participants expressing disbelief in the task were excluded and its relevance to study goals and biases.

15. Provide context for variations in model fit, especially for participants with lower predicted reciprocation values.

16. Highlight practical implications of observed model differences, even if not statistically significant.

17. Discuss limitations of not correcting for multiple tests and consider reporting effect sizes and confidence intervals.

18. The last paragraph in the "Task" subsection may benefit from additional explanations or a specific diagram to visualize the trial distribution and the relationship between investments and endowments.

19. Provide a more explicit rationale for selecting trust and investment as fixed effects. Explain the significance of using a sum-to-zero contrast scheme and clarify the inclusion/exclusion of random effects like the trust-investment interaction.

20. Briefly explain why specific statistical methods were chosen for analysis to aid reader understanding.

Results

20. Use concise language to describe modeling procedures for easier comprehension.

21. Provide context for variations in model fit, especially for participants with lower predicted reciprocation values.

22. Clarify practical implications of observed model differences, even if not statistically significant.

23. Discuss limitations of not correcting for multiple tests and consider reporting effect sizes and confidence intervals.

24. Rearrange references to Appendices B and C and consider adding more detail to Appendix B.

Discussion

25. Present results thematically rather than enumerating them for improved readability.

26. Strengthen integration of existing literature by explicitly referencing relevant studies and theories.

27. Restate the study's hypotheses and summarize alignment with or deviations from them. Explore practical implications of individual differences.

28. Organize a dedicated section on future research directions, outlining key questions and how future research might address them.

29. Discuss broader findings' implications for real-world social decision-making scenarios.

Reviewer #2: The authors of this paper present an experiment aimed at understanding how individuals reciprocate trust in ambiguous situations within the trust game. Through computational modeling, the study reveals that individuals frequently rely on social heuristics grounded in past interactions when the degree of trust is clear.

While the paper has the potential to significantly contribute to the existing literature, there are several areas where it could be improved:

- "This raises the possibility that people might use these ambiguous situations to make strategic decisions that maximize both self-interest and self-image"; I think it is more precise to say "...while preserving their self-image."

- "we can about how people resolve": I think there is a typo here.

- In discussing motivations for reciprocating trust, the authors might benefit from mentioning also the recent "moral preferences hypothesis." This theory suggests that many prosocial behaviors, including trustworthiness, emanate from an inherent moral desire to act rightly, sometimes beyond mere distributional preferences. As a review article, the authors could consult: https://royalsocietypublishing.org/doi/full/10.1098/rsif.2020.0880. This may necessitate considerable revisions, especially given the paper's reliance on modeling reciprocation using inequity-aversion.

- The fairness norm model employed by the authors bears resemblance to the model by Kessler and Leider: https://pubsonline.informs.org/doi/abs/10.1287/mnsc.1110.1341. For a more comprehensive overview and to draw clearer distinctions from prior work, I recommend consulting this review article on utility functions focused on formalizing moral preferences: https://www.aeaweb.org/articles?id=10.1257/jel.20221613

- "Here, given that M (the multiplier) is usually fixed at 4 in these games": Typically, the multiplier is set at 3, and this seems to be the case in the provided formula as well. So, perhaps this is just a typo.

- It is not clear why the authors name the fourth strategy as "heuristic". I have the feeling that also the third strategy is a heuristics. It seems similar to what Rand and colleagues call "social heuristic" (e.g., https://www.nature.com/articles/ncomms4677).

- Referring to the trust game as the "Investment Game" to prevent behavioral bias could be counterproductive, as the term "investment" might introduce its own set of biases.

- The clarity and quality of the figures should be enhanced, as they currently appear blurry.

- A "Limitations" subsection would be a valuable addition. It is important to acknowledge inherent limitations in experimental research. For instance, while the number of trials might offer high power, the relatively small sample size hinders in-depth exploration of participant heterogeneity.

6. PLOS authors have the option to publish the peer review history of their article (what does this mean?). If published, this will include your full peer review and any attached files.

Reviewer #1: No

Reviewer #2: No

---

## [Author Response · Author response to Decision Letter 0]

8 Jan 2024

Manuscript PONE-D-23-24513

Response to Reviewers

Dear Ms. Recto,

Thank you for giving us the opportunity to submit a revised draft of the manuscript “Reciprocity in Ambiguous Situations: Default Psychological Strategies Underlying Ambiguity-Resolution in Moral Decision-Making” for publication in PLOS One. We are appreciative of the thought and effort which both you and the reviewers have put into providing us with suggestions to improve the quality of our manuscript. In this document, you will find point-by-point responses to these suggestions, most of which have been incorporated into the revised draft of the manuscript. All page numbers refer to the revised manuscript file without tracked changes. 

Editor’s Comments to the Authors:

1. Ensure that your manuscript meets PLOS ONE's style requirements, including those for file naming. The PLOS ONE style templates can be found at 

Author Response: The current manuscript has been adapted to meet PLOS ONE’s style requirements.

2. Thank you for submitting the above manuscript to PLOS ONE. During our internal evaluation of the manuscript, we found significant text overlap between your submission and previous work in the [introduction, conclusion, etc.].

Please revise the manuscript to rephrase the duplicated text, cite your sources, and provide details as to how the current manuscript advances on previous work. Please note that further consideration is dependent on the submission of a manuscript that addresses these concerns about the overlap in text with published work.

[If the overlap is with the authors’ own works: Moreover, upon submission, authors must confirm that the manuscript, or any related manuscript, is not currently under consideration or accepted elsewhere. If related work has been submitted to PLOS ONE or elsewhere, authors must include a copy with the submitted article. Reviewers will be asked to comment on the overlap between related submissions (http://journals.plos.org/plosone/s/submission-guidelines#loc-related-manuscripts).]

We will carefully review your manuscript upon resubmission and further consideration of the manuscript is dependent on the text overlap being addressed in full. Please ensure that your revision is thorough as failure to address the concerns to our satisfaction may result in your submission not being considered further.

Author Response: The similarities noted are between the submitted article and the first authors Masters thesis, from which the current submission has been developed. This submission has not been published elsewhere, nor is it currently under review at any venue. 

Author Response: These accession numbers are now provided in the Methods section.

Author Response: These changes have been made.

Reviewers' Responses to Questions:

1. Is the manuscript technically sound, and do the data support the conclusions?

Reviewer #1: Partly

Reviewer #2: Yes

2. Has the statistical analysis been performed appropriately and rigorously?

Reviewer #1: I Don't Know

Reviewer #2: Yes

3. Have the authors made all data underlying the findings in their manuscript fully available?

Reviewer #1: No

Reviewer #2: Yes

4. Is the manuscript presented in an intelligible fashion and written in standard English?

Reviewer #1: Yes

Reviewer #2: Yes

5. Review Comments to the Author

Reviewer #1: Review of Manuscript

The paper presents a novel exploration of ambiguity in reciprocity decisions, employing a computational modeling approach. While the study provides valuable insights into how individuals handle uncertainty, several areas can be refined. The entire paper, especially the Methods section, would benefit from more explicit explanations. It's important to delve deeper into participant selection criteria and elaborate on experimental design choices for better reader understanding. In the Results section, enhancing the systematic presentation of findings and aligning them with the study's objectives would enhance clarity. The Discussion section could be improved through more extensive comparisons with existing literature and stronger connections to the study's objectives and hypotheses. Despite these areas for improvement, this research lays a promising foundation for further investigating ambiguity in social decision-making. It underscores the importance of future studies considering individual differences in decision strategies. Below, you will find my comments for each section of the manuscript.

Author Response: Thank you! Please see our responses to your helpful suggestions below. 

Introduction

1. Explicitly state the literature gap the study addresses. Identify unanswered questions regarding ambiguity in trust to set a focused research context. What unanswered questions or uncertainties about ambiguity in trust exist in the current research landscape? Highlight the study's key contributions to establish clear expectations for readers.

Author Response: We thank the reviewer for this helpful suggestion, and have now reformulated several sentences to make more explicit the key unanswered questions this study addresses:

‘Since no previous studies have investigated how people reciprocate ambiguous trust, we do not yet have an empirical answer to this question. In fact, it is possible that people never reciprocate ambiguous trust; if this is true, it would undermine the conclusions of previous laboratory studies which studied reciprocity under conditions of certainty, since these studies often show that the vast majority of people reciprocate trust. Consequently, in this study we will determine if people reciprocate trust when it is ambiguous. More specifically, since reciprocity is determined by the trust which has been bestowed, we will identify how this unknown bestowed trust is imputed in order to make reciprocation decisions.’ (page 3, paragraph 3).

In order to make this section flow better, we have separated this paragraph into two paragraphs.

2. Consider relocating equations, like Equation 3, to the Methods section. This streamlines the introduction, making it more accessible. Provide concise explanations of specialized terminology and equations for broader readership. Add transitional sentences to link the introduction to subsequent sections.

Author Response: We agree with this suggestion and have now relocated Equation 3 to the Methods section. We also now provide an additional detail to clarify Equation 1:

‘Specifically, a Δ of 0 would predict that the participant never returns anything, a Δ of 0.5 would predict that the participant always returns enough so that their payout and the investor’s payout are equal, and a Δ of 1 would predict that the participant always returns everything to the investor.’ (page 7, paragraph 2). 

Further, we now provide a definition of endowment as indeed this term was not explicitly defined: 

‘The term Endowment refers to the money provided for use in the game.’ (page 4, paragraph 2). 

Additionally, we have included several transition sentences to the ‘Current Study’ section: 

‘By making the endowment ambiguous, we can infer how people resolve the resulting ambiguity about the trust their partner has bestowed upon them.’ (page 5, paragraph 3)

’Equation 2 explains how we propose to measure ambiguity-resolution in the current study, and we will next focus on predicting this ambiguity-resolution.’ (page 7, paragraph 3).

3. Provide a concise explanation of the ARS model and its relevance. Establish a clear link between the Offer Game discussion and the ARS model introduction.

Author Response: We now include additional information to help define the ARS model and its relevance, as suggested. Specifically, we now state: 

‘Thus, the ARS model incorporates all of the potential explanations for ambiguity-resolution behavior in that it captures financially advantageous choices via the LAT strategy, captures benefit of the doubt choices via the HAT strategy, captures previous experience-informed choices via the MT strategy, and captures larger-is-more choices via the HR strategy.’ (page 8, paragraph 2).

4. Elaborate on the "veil of ambiguity" concept's relevance and implications for the research. Explain its connection to the Trust Game and study objectives.

Author Response: While we do see the value in expounding upon the relevance and implications, we do not think that elaborating on the implications of the ‘veil of ambiguity’ (we did call this the ‘veil of ignorance’ but it has been changed since it different to Rawls’ veil of ignorance, and we want to avoid any misunderstanding) is necessary, particularly since the results suggest that it does not describe the ambiguity-resolution process. In fact, we believe that including a specific term to refer to this tendency may be counterproductive since it may lead to a focus on this proposed pattern of behavior which was not ultimately supported by the data. As such, we have reformulated this sentence: 

‘Responders may be taking advantage of the ambiguity, pretending that offers are fair even when they think they are actually unfair. Since unfair offers might compel the responder to reject these offers, pretending that offers are fair can circumvent this compulsion and thereby maximize the responder’s payout.’ (page 4, paragraph 2).

5. Simplify complex sentences and explanations for better readability and understanding.

Author Response: We thank the reviewer for calling attention to the somewhat densely formulated text. To improve the readability and facilitate easier understanding, we have extensively edited the manuscript to better accomplish these goals.

6. Consider using diagrams or graphs to illustrate how ambiguity is introduced in the Hidden Endowment Trust Game.

Author Response: This is a very helpful suggestion to aid in the comprehension of our task. We have created a new figure – Figure 1 (page 6) – which shows the implications of the potential endowments in both the unambiguous and ambiguous conditions. Thus, the reader should now be able to more easily grasp how ambiguity was induced in the HETG and why reciprocity requires resolving this ambiguity.

Methods

7. Briefly explain the importance of preregistration for research transparency and bias reduction.

Author Response: We thank the reviewer for drawing attention to this important point. We have included a few sentences which speak to the importance of preregistration: 

‘Preregistration is important for increasing transparency in the research process, as it clarifies distinctions between confirmatory and exploratory analyses prior to the collection of data. Thus, confirmatory results reported by preregistered studies can be trusted to not be merely exploratory results, which ensures that the False Positive Rate – or Type II error – of these results is accurately reported.’ (page 9, paragraph 3).

8. Clarify the exclusion of left-handed participants and those with non-removable metal objects. Address whether these were the sole exclusion criteria for fMRI participants. Justify excluding local Behavioral Science Research Master students and discuss its impact on generalizability.

Author Response: We excluded left-handed participants and those with non-removable metal objects as part of this study was conducted using functional magnetic resonance imaging (fMRI). These are very common exclusion criteria for fMRI studies. Left-handed participants – who constitute 11% of the population – are excluded in order to reduce variability in neuroimaging results which arise from differences in functional organization related to handedness. The presence of non-removable metal objects produces a hazard for participants’ safety, since these objects can be moved by the strong magnetic fields in the fMRI scanner. Behavioral Science Research Master students from Radboud University were excluded since this group had foreknowledge about the study’s procedure and aims which could have led to clear demand effects. Of note, the latter were a very small and specific group of potential participants (~40 in total), hence we do not believe that excluding these potential participants had an effect in terms of generalizability.

9. Provide context for the sequential Trust Game and HETG. Explain its influence on the experimental design.

Author Response: We currently state that:

‘participants always played the unambiguous condition before the ambiguous condition in order to ensure they had previous experience upon which to base their decision-making.’ 

We will augment this explanation to further clarify this motivation by adding the text: 

‘Conditions were played sequentially to remove the possibility of learning effects: if conditions were intermingled, participants would learn more about how trusting participants are on a trial-by-trial basis. Thus, this learned information has the potential to influence ambiguity resolution on a trial-by-trial basis – to avoid these contamination effects, we opted to have the Trust Game and HETG played sequentially.’ (page 13, paragraph 1).

10. Describe participant training or instructions for the Trust Game and HETG to shed light on decision-making processes.

Author Response: We have now included the following text:

‘Participants were given clear instructions for the Trust Game, and then played a handful of practice trials to ensure that they understood how payouts were determined for both players based on the 1) Investment amount and 2) the Returned amount.’ (page 11, paragraph 1).

11. Provide a detailed explanation of the payment structure, outlining the specific criteria and mechanisms through which participants could earn the additional €16 based on their decisions and Player A's choices.

Author Response: The specific mechanism of bonus payment calculation was not provided to participants directly, and hence did not impact their decision process. However, for full clarification, we have added the procedure for determining final payment to the appendix. 

12. Elaborate on the reasoning behind programming the Investor's choices. This additional information will enhance readers' comprehension of the programming's purpose and its importance in the study.

Author Response: We have now included the following text: 

‘We programmed Investor’s choices so that we would have a complete set of trials which was diverse enough to ensure that we could recover our free parameters.’ (page 13 paragraph 1).

13. Explain how the measures address research questions and why computational modeling suits ambiguity-resolution.

Author Response: We have clarified this important point by adding the following text: 

“Computational models are used to capture the motives which could be used to guide decision-making, which make them a valuable tool for determining if people differ in which motives guide their decision-making. Use of these computational approaches then allows us to test if these hypothesized motivations accurately reflect their actual decision behavior.” (page 8, paragraph 2).

14. Clarify why participants expressing disbelief in the task were excluded and its relevance to study goals and biases.

Author Response: We have now included the following text: 

‘Participants who expressed disbelief in the task were excluded since these participants’ decisions were unlikely to reflect their actual social preferences, undermining the validity of the conclusions which can be drawn from their data.’ (page 10, paragraph 3). 

15. Provide context for variations in model fit, especially for participants with lower predicted reciprocation values.

Author Response: We have now included the following sentences in the results section:

‘Similar to previous studies, these variations in model fit reflect the notion that reciprocation behavior in the Trust Game is difficult to model for all participants in the same way. Specifically, these variations arise since divisions can be influenced by the multiplied investment, particularly when a partner is untrusting. Thus, for a small minority of participants, the model overpredicts reciprocation amounts.’ (page 16, paragraph 2).

16. Highlight practical implications of observed model differences, even if not statistically significant.

Author Response: We now highlight this in the discussion section, under the ‘Reflection on Null Findings’ subsection.

17. Discuss limitations of not correcting for multiple tests and consider reporting effect sizes and confidence intervals.

Author Response: While we of course agree with the reviewer’s point about the general inadvisability of not correcting for multiple tests, we respectfully disagree that this is a limitation given the current study design and approach. We currently state that: 

‘Our justification for not correcting for multiple tests was that one single-strategy model would fit all participants better than the other three single-strategy models so, ultimately, we would be testing the ARS model against only the best-fitting single-strategy model.’ 

Therefore, we argue here that our goal was specifically to test for the presence of individual differences in strategy adoption - if the model which captures multiple strategies performs better than the best model which only captures a single strategy, then there are by definition individual differences in strategies. 

18. The last paragraph in the "Task" subsection may benefit from additional explanations or a specific diagram to visualize the trial distribution and the relationship between investments and endowments.

Author Response: We agree, and believe that the newly added Figure 1 helps illustrate the relationship between Investments and Endowments.

19. Provide a more explicit rationale for selecting trust and investment as fixed effects. Explain the significance of using a sum-to-zero contrast scheme and clarify the inclusion/exclusion of random effects like the trust-investment interaction.

Author Response: We chose trust, investment, and trust-investment as fixed effects because they are the independent variables in our experiment. We chose a sum-to-zero contrast scheme (and centered continuous predictors) because sum-to-zero contrasts make the condition effects uncorrelated with each other – this explanation is now provided alongside our indication of our chosen contrast. Finally, we indicate that we specified all fixed effects as random effects, but this led to a singularity warning so we re-estimated the model without the random effect for trust-investment. Singularity warnings imply that multiple terms in the model explain the same variance – in this case, it seems that the fixed-and-random effects for trust-investment were explaining the same variance, when accounting for random effects of trust and investment separately. In other words, there were no individual differences in the trust-investment slope. This too, is now clarified.

20. Briefly explain why specific statistical methods were chosen for analysis to aid reader understanding.

Author Response: We have now added text to more clearly explain our motivation in using computational modeling to measure ambiguity resolution, and we explain our motivation for using computational modeling to study ambiguity-resolution strategies in both the introduction, results, and Conclusion. We have added the following sentence to the end of the Mixed Effects Modeling subsection: 

“In order to identify group-level trends in reciprocation behavior as a function of bestowed trust while controlling for individual differences, we adopted a mixed effects modeling approach” (page 15, paragraph 1).

Results

21. Use concise language to describe modeling procedures for easier comprehension.

Author Response: While we do recognize that the modeling procedures are described in complex terms which may be difficult for some readers without sufficient computational modelling experience to understand, we do deem it necessary to describe the modeling procedures as unambiguously as possible to enable other researchers to potentially replicate the entire analysis pathway. As indicated in our responses to previous points of the reviewer, we have endeavored to simplify the language throughout, in particular in the conceptual sections of the paper, and we believe this should suffice for general readers to broadly understand the methods applied, while readers with greater computational backgrounds are provided with the detailed formulations as currently described in the results section.

22. Provide context for variations in model fit, especially for participants with lower predicted reciprocation values.

Author Response: We have now added clarification for this variability under the ‘Limitations’ section of the Conclusion: 

‘This is a common finding when modeling reciprocity in the context of the Trust Game since division norms can be applied to all of the money in the game – as in our model – but, alternatively, to the multiplied investment alone or a combination of the two. Thus, this heterogeneity is not accounted for in the model, which explains why a minority of participants’ data was not explained at lower reciprocation values. Pragmatism notwithstanding, this calls into question the ability to make reliable reverse inferences for this minority of participants whose data was overpredicted at low trust values and underpredicted at high trust values.’ (page 19, paragraph 2). 

23. Clarify practical implications of observed model differences, even if not statistically significant.

Author Response: We discuss this in the Conclusion section, under the ‘Reflection on Null Findings’ subsection (page 23, paragraph 3).

24. Discuss limitations of not correcting for multiple tests and consider reporting effect sizes and confidence intervals.

Author Response: See point 17 above for our response regarding multiple-test corrections. We report the effect size of model differences between computational models, in order to provide future studies with the necessary information for power analyses (i.e. to find significant individual differences). We do not report the effect size from the mixed effects modeling analysis because a correlation would be the ideal way to characterize the effect of trust on reciprocity, rather than the raw effect of the condition-over-condition differences. We thank the reviewer for calling attention to this, as well as the confidence intervals which we will also omit since this would be redundant information given that we report the Regression Coefficients as well as the Standard Error.

25. Rearrange references to Appendices B and C and consider adding more detail to Appendix B.

Author Response: Thank you for pointing this out, the appendices are now in the correct order. We have removed what was formerly Appendix B: since the grouping of the correct parameter space was not used in our results, we see no reason to have included the a priori specified one.

Conclusion

26. Present results thematically rather than enumerating them for improved readability.

Author Response: These results are now presented `Conclusions > Reflection on Null Results > Limitations > Strengths of this Study > Future Directions`, and we agree that this provides greater readability. 

27. Strengthen integration of existing literature by explicitly referencing relevant studies and theories.

Author Response: This is a helpful suggestion, and the section on the Reflection on Null Results now explicitly connects the LAT, HAT, and HR strategies – which did not characterize the ambiguity-resolution strategies that people actually adopted – to relevant psychological theories which would have predicted these respective patterns of behavior.

28. Restate the study's hypotheses and summarize alignment with or deviations from them. Explore practical implications of individual differences.

Author Response: These are now included under the Conclusions and Reflection on Null Results sections. 

29. Organize a dedicated section on future research directions, outlining key questions and how future research might address them.

Author Response: We have now added this useful section, which focuses on the potential to study decision-making under ambiguity in more realistic scenarios – namely, repeated interactions and scenarios with additional information which might be relevant to trust assessments.

30. Discuss broader findings' implications for real-world social decision-making scenarios.

Author Response: We discuss these implications in the section outlined above (point 28), as well as in the Limitations section, in which we reflect on the fact that – in real-world situations where people usually have at least some information about the person they are interacting with – it is unlikely that default strategies to resolve ambiguity strongly influence reciprocity decisions in a self-interested direction. 

Reviewer #2: The authors of this paper present an experiment aimed at understanding how individuals reciprocate trust in ambiguous situations within the trust game. Through computational modeling, the study reveals that individuals frequently rely on social heuristics grounded in past interactions when the degree of trust is clear.

While the paper has the potential to significantly contribute to the existing literature, there are several areas where it could be improved:

Author Response: Thank you! Please see our responses to your helpful suggestions below. 

1. "This raises the possibility that people might use these ambiguous situations to make strategic decisions that maximize both self-interest and self-image"; I think it is more precise to say "...while preserving their self-image."

Author Response: We agree with this helpful point, and the manuscript has now been changed to reflect this.

2. "we can about how people resolve": I think there is a typo here.

Author Response: We thank the reviewer for their attention to detail, and have now corrected this typo. 

3. In discussing motivations for reciprocating trust, the authors might benefit from mentioning also the recent "moral preferences hypothesis." This theory suggests that many prosocial behaviors, including trustworthiness, emanate from an inherent moral desire to act rightly, sometimes beyond mere distributional preferences. As a review article, the authors could consult: https://royalsocietypublishing.org/doi/full/10.1098/rsif.2020.0880. This may necessitate considerable revisions, especially given the paper's reliance on modeling reciprocation using inequity-aversion.

Author Response: Our belief about the implications of Capraro and Perc’s argument (2021) is that it does not necessarily contradict our reliance on modeling reciprocation using inequity-aversion. As you indicate, the moral preferences hypothesis pertains to a moral desire to act correctly – reflecting intentions underlying distributional preferences. Our hypothesis is that inequity-aversion (or guilt-aversion which, as we state, is behaviorally captured in this paradigm with the inequity-aversion model) is a norm which people reference in order to assess if they have indeed acted in a morally correct manner. In fact, the studies by van Baar and colleagues (2019, 2020; which indicate that different norms guide reciprocity for different people) overtly argue for the same conclusion; however, our design does not treat reciprocity as anything more than a behavioral distribution from which we can infer how people resolve ambiguity. Nonetheless, the reviewer’s suggestion represents a powerful frame through which to interpret our finding that most people use their previous experience to resolve ambiguity. Namely, using this framework our results indicate that – just like reciprocity – ambiguity-resolution is motivated by a desire to act in a morally correct manner (page 18, paragraph 2).

4. The fairness norm model employed by the authors bears resemblance to the model by Kessler and Leider: https://pubsonline.informs.org/doi/abs/10.1287/mnsc.1110.1341. For a more comprehensive overview and to draw clearer distinctions from prior work, I recommend consulting this review article on utility functions focused on formalizing moral preferences: https://www.aeaweb.org/articles?id=10.1257/jel.20221613

Author Response: The review article helpfully suggested by the reviewer directly mentions both of the inequality-aversion models papers that our model is based on (Bolton & Ockenfels, 2000 and Schmidt & Fehr, 1999). The first paper was already cited, and we now cite both in the Introduction when speaking about inequality-aversion; additionally, we make mention of this review paper in the following sentence: 

‘Informed by previous studies which have formally modeled moral preferences in utility equations (for a review of this research, see Capraro, Halpern, & Perc, 2022), the current study has formalized a behavioral model which captures inequality-aversion without a utility term. This approach enables us to determine how participants resolved ambiguity, since we can solve for the unknown endowment when we know their division norm.’ (page 7, paragraph 2).

We should note that we do not discuss these papers in great detail as, strictly speaking, we do not employ a utility model in Equation 1.

5. "Here, given that M (the multiplier) is usually fixed at 4 in these games": Typically, the multiplier is set at 3, and this seems to be the case in the provided formula as well. So, perhaps this is just a typo.

Author Response: To clarify, this is not a typo – the algebraic expression is rearranged. The generic equation to encapsulate all of the money in the game is M*I + (E – I). Here, we see that it is possible to rearrange this equation to M*I – I + E which enables us to factor I out of M*I and - I which means we can simplify this expression to (M – 1)*I + E. Thus, since the multiplier, M, is equal to 4 in our experiment, we express this term as 3I + E. 

6. It is not clear why the authors name the fourth strategy as "heuristic". I have the feeling that also the third strategy is a heuristics. It seems similar to what Rand and colleagues call "social heuristic" (e.g., https://www.nature.com/articles/ncomms4677).

Author Response: We thank the reviewer for calling attention to this point of uncertainty. We have now amended the manuscript to include the following statement when discussing the fourth strategy: 

‘We termed the fourth strategy as a heuristic strategy, as those that use it report employing a straightforward rule that is basically ‘lower investments equal lower trust’; thus, a simple heuristic is applied to the amount that has invested to resolve ambiguity.’ (page 8, paragraph 1). 

For the third strategy, we felt that ‘heuristic’ was not the best descriptor as those that use it report actively trying to remember what their previous knowledge about this partner was. Thus, these people were attempting to resolve ambiguity via direct previous experience. While we understand that the term ‘heuristic’ itself is somewhat ill-defined, we believe the essence of this third strategy distinguishes it from the fourth outlined above.

7. Referring to the trust game as the "Investment Game" to prevent behavioral bias could be counterproductive, as the term "investment" might introduce its own set of biases.

Author Response: This is a fair point, and we have now amended the manuscript to acknowledge these potential implications: ‘Our rationale for this choice was that by framing the decisions as ‘investments’ we might avoid the possibility that reciprocity was a consequence of a social desirability effect.’ (page 11 paragraph 1).

8. The clarity and quality of the figures should be enhanced, as they currently appear blurry.

Author Response: When downloading the files uploaded in the original submission, we could not identify any figures which appeared blurry nor did we see any way in which the quality could be enhanced. As such, no changes have been made.

9. A "Limitations" subsection would be a valuable addition. It is important to acknowledge inherent limitations in experimental research. For instance, while the number of trials might offer high power, the relatively small sample size hinders in-depth exploration of participant heterogeneity.

Author Response: We agree, and have now added a section to the Conclusion outlining more clearly the study limitations (page 22, paragraphs 2-3).

---

## [Decision Letter · Decision Letter 1]

2 Feb 2024

PONE-D-23-24513R1Reciprocity in Ambiguous Situations: Default Psychological Strategies Underlying Ambiguity-Resolution in Moral Decision-MakingPLOS ONE

Dear Dr. Galvan,

Thank you for submitting your manuscript to PLOS ONE. After careful consideration, we feel that it has merit but does not fully meet PLOS ONE’s publication criteria as it currently stands. Therefore, we invite you to submit a revised version of the manuscript that addresses the points raised during the review process.

There are still several comments by one of the reviewers that should be addressed and would improve the manuscript. 

We look forward to receiving your revised manuscript.

Kind regards,

Inon Zuckerman

Academic Editor

PLOS ONE

Journal Requirements:

Additional Editor Comments:

Following the review of the first revision there are several comments that needs to be addressed.

Reviewers' comments:

Reviewer's Responses to Questions

**Comments to the Author**

1. If the authors have adequately addressed your comments raised in a previous round of review and you feel that this manuscript is now acceptable for publication, you may indicate that here to bypass the “Comments to the Author” section, enter your conflict of interest statement in the “Confidential to Editor” section, and submit your "Accept" recommendation.

Reviewer #1: (No Response)

Reviewer #2: All comments have been addressed

2. Is the manuscript technically sound, and do the data support the conclusions?

Reviewer #1: Yes

Reviewer #2: Yes

3. Has the statistical analysis been performed appropriately and rigorously? 

Reviewer #1: Yes

Reviewer #2: Yes

4. Have the authors made all data underlying the findings in their manuscript fully available?

Reviewer #1: Yes

Reviewer #2: Yes

5. Is the manuscript presented in an intelligible fashion and written in standard English?

Reviewer #1: Yes

Reviewer #2: Yes

6. Review Comments to the Author

Reviewer #1: Although the authors have made an effort to address most of my concerns, further refinement could enhance the manuscript's comprehensiveness and robustness. Specifically:

1. The manuscript would benefit from deeper discussions on behavioral trends and decision-making in ambiguous situations, with a stronger connection to psychological or moral theories. These sections, notably around pages 20 and 21, could be strengthened with additional literature citations.

For example, the papers "Attributional Ambiguity Reduces Charitable Giving by Relaxing Social Norms" and "Social Game Theory: Preferences, Perceptions, and Choices" are examples of relevant literature that can enrich these discussions. While the cited papers may provide a starting point, authors should consider a broader range of literature for a richer theoretical foundation.

2. Figures 1 and 2 should be improved for better legibility. This includes enhancing font size and image clarity. Additionally, the description of Figure 1 and the table in Appendix B requires more detailed explanation to ensure that readers can understand the experimental design and results without ambiguity. The manuscript currently has two appendices labeled "Appendix B" which could lead to confusion and should be rectified.

3. The limitations section should reflect the use of a student population and suggest the need for a more diverse sample in future studies.

4. For enhanced clarity it is advised that the figure currently labeled as "Appendix B," which depicts different strategies, should be more comprehensively explained within the text. Specifically, this explanation should clearly connect to the detailed description of the parameter space provided on page 15. Such integration will help readers understand the significance of the strategies in the context of the study's outcomes and the computational modeling approach the authors have employed.

5. Ensure the accuracy of the reference list, particularly for citations like Capraro et al., to maintain consistency and correctness.

Reviewer #2: The authors have addressed all my comments. I have only noticed that many references are missing from the new reference list. I suggest the authors to make sure that there is a matching between in-text references and bibliography.

7. PLOS authors have the option to publish the peer review history of their article (what does this mean?). If published, this will include your full peer review and any attached files.

Reviewer #1: No

Reviewer #2: No

---

## [Author Response · Author response to Decision Letter 1]

26 Feb 2024

Rebuttal Letter

We would like to take the opportunity to thank both reviewers for their constructive suggestions to improve our manuscript. We deeply appreciate the opportunity to submit another revision of our paper for your consideration. Below, we respond to the comments made by the reviewers and list the subsequent changes made in the paper. The page and paragraph numbers refer to those in the manuscript with track changes.

Reviewer concerns

Reviewer 1

Although the authors have made an effort to address most of my concerns, further refinement could enhance the manuscript's comprehensiveness and robustness. Specifically:

1. The manuscript would benefit from deeper discussions on behavioral trends and decision-making in ambiguous situations, with a stronger connection to psychological or moral theories. These sections, notably around pages 20 and 21, could be strengthened with additional literature citations. For example, the papers "Attributional Ambiguity Reduces Charitable Giving by Relaxing Social Norms" and "Social Game Theory: Preferences, Perceptions, and Choices" are examples of relevant literature that can enrich these discussions. While the cited papers may provide a starting point, authors should consider a broader range of literature for a richer theoretical foundation.

Our response: 

We would like to thank the reviewer for bringing attention to this opportunity to improve our paper and, in particular, the helpful direction provided with the two recommended papers. We have now added the following paragraph to the Discussion.

“In the context of the larger body of work on social decision-making under ambiguity, our results are somewhat anomalous: many researchers find that decision-making under ambiguity is less prosocial. Therefore, we must consider what specifically about the HETG distinguishes it from the tasks used in these previous studies. For example, in both the Public Goods Game (PGG) and Prisoner’s Dilemma (PD), wherein participants must make the decision to cooperate or defect without knowing their partner’s decision or past behavior, it has been consistently reported that participants cooperate less often than they defect (Gächter et al., 2017; Flood & Drescher, 1950; Zelmer, 2003). In a version of the PGG wherein the group must cooperate to reach a threshold of contributions to the community pot or otherwise lose all their money, participants cooperate more; however when ambiguity is introduced about what this threshold is, participants cooperate much less (Dannenberg, Lange, & Sturm, 2014). In this regard, it is possible that the specific ambiguity which is present in the HETG has a diverging effect because behavioral distrust is incentive-incompatible. In other words, participants in the HETG know that, regardless of any beliefs that they might have about how trusting investors are in general, their partners will invest more if they trust the participant more. Indeed, previous research on the PD suggests that perceptions of morality are insensitive to outcomes (Krueger et al., 2020). In his light the results from the HETG may therefore be attributed to a general aversion to behaving in what may be regarded as an immoral way simply because the context of the HETG does not offer sufficient cover to morally justify behaving selfishly. Concordantly, Tho Pesch and Dana (2024) find that charitable giving decreases when participants must make a choice about which charity to donate to before making the giving decision, rather than just making only the giving decision. The obfuscation of the charitable choice by simply providing attributes for comparison has been referred to as attributional responsibility (Snyder et al., 1979). In regard to the HETG, the lack of attributional ambiguity – or the inability to, perhaps, attribute ostensibly selfish behavior to anything other than a response to distrust, which was unmerited based on their previous experience, may be responsible for this atypical pattern of behavior.” (page 21, paragraph 1)

2. Figures 1 and 2 should be improved for better legibility. This includes enhancing font size and image clarity. Additionally, the description of Figure 1 and the table in Appendix B requires more detailed explanation to ensure that readers can understand the experimental design and results without ambiguity. The manuscript currently has two appendices labeled "Appendix B" which could lead to confusion and should be rectified.

Our response:

We thank the reviewer for bringing attention these discrepancies. With regard to the image quality of Figures 1 and 2, we would like to reiterate that the pdf printout of the files does indeed make these images appear pixelated, since it significantly magnifies these images in order to present them at the size of the a page. However, when downloading the images one can see that the images are of substantially better quality when viewed at their original size. With regard to the text size of Figure 1, we believe that it is large enough for most to read while being small enough to keep this already large image from taking up more space on a page. Concerning the text size of Figure 2, we agree with the reviewer and have therefore made a new version of this image which is larger so as to improve legibility while preserving the proportions of each screen. 

Concerning the ordering of the Appendices, we would like to express our appreciation for the reviewer’s attention to detail. During the submission of our revision, we included the previously uploaded Figures so that the reviewers could more easily reference which Appendices were being removed – however, this has clearly led to some confusion which is our fault. The only Appendix which remains and contains an image is Appendix C – showing the Q-Q plot of the behavioral reciprocity model. The former Appendix B was removed in the previous revision, which we acknowledged in point 25 of our response to your previous review: ‘We have removed what was formerly Appendix B: since the grouping of the correct parameter space was not used in our results, we see no reason to have included the a priori specified one.’ We further address this point below in our response to point 4.

3. The limitations section should reflect the use of a student population and suggest the need for a more diverse sample in future studies.

Our response: 

We agree, and have now added a new paragraph outlining the limitations of using a student population, as well as the limitations associated with our use of a WEIRD population more generally. 

“A further limitation of the current study is the population which the sample was drawn from. Since our sample was comprised nearly exclusively of University students, the conclusions we can draw from our data are inherently constrained by the fact that we have sampled a portion of the adult population which is rather young, often of a relatively high SES, and share a similar educational background. For instance, respondents’ propensity to use an accuracy-oriented approach to resolve ambiguity may not be characteristic of a broader population who may be less idealistic and who, being generally less privileged, may be more self-interested. In a similar regard, the current study was conducted on a WEIRD population: WEIRD populations – which are Westernized, Educated, Industrialized, Rich, and Democratic – comprise a disproportionate majority of research on human psychology (Arnett, 2008; Rad, Martingano, & Ginges, 2018). The current study falls into this pattern: convenience samples are often used in the first stages of research, but further research into human decision-making under ambiguity must be done with a more diverse sample than we utilize here.“ (page 25, paragraph 2)

4. For enhanced clarity it is advised that the figure currently labeled as "Appendix B," which depicts different strategies, should be more comprehensively explained within the text. Specifically, this explanation should clearly connect to the detailed description of the parameter space provided on page 15. Such integration will help readers understand the significance of the strategies in the context of the study's outcomes and the computational modeling approach the authors have employed.

Our response: 

We agree with the reviewer that it would help to explain what the meaning of the free parameters in our computational model. However, as we have noted previously, including the plotting of the parameter space is not merited given that we do not use the computational model to group participants because the computational model is not significant. Therefore, we believe that the most appropriate place for this discussion is in the Methods section, where we introduce the ARS model (Equation 4). Therefore, we have now added the following short paragraph after Equation 4:

“The ARS model captures individual differences via its free parameters. It captures participants who make decisions using the heuristic strategy with its ψ parameter such that high ψ values make it important that the trust implied by a given endowment agrees with the trust implied by the investment alone (i.e. 1 – abs(I/E) – (I/10)). Inversely, low ψ values make it important that the trust implied by a given endowment agrees with the participant’s default assumed trustingness – or τ. Therefore, when ψ is high, the model predicts that participants will follow the HR strategy. When ψ is low on the other hand, τ represents the strategy: values of τ close to 0 predict the LAT strategy, values of τ close to 1 predict the HAT strategy, and values of τ close to 0.5 predict the MT strategy.“ (page 16, paragraph 2)

5. Ensure the accuracy of the reference list, particularly for citations like Capraro et al., to maintain consistency and correctness.

Our response:

Thank you for pointing out these missing references. It seems that when performing the final step of proof-reading, we neglected to ensure that the new references had been included in the Reference List. We apologize profusely for allowing this oversight to happen. The missing references and new references are now included in the new submission.

Reviewer 2

The authors have addressed all my comments. I have only noticed that many references are missing from the new reference list. I suggest the authors to make sure that there is a matching between in-text references and bibliography.

Our response:

We would like to thank the reviewer for pointing out these missing references. As mentioned above, we simply neglected to ensure that the new references had been included in the reference list. We would like to express our apologies for allowing this oversight to occur. The missing and new references are now included in the new submission.

Editor concerns

Our response:

We thank the editor for bringing attention to accuracy of our reference list. Using our reference manager, Zotero, we have ensured that no references are made to retracted articles. Our reference article is now updated to include all citations of works in the manuscript and any further changes to our reference list are explicitly mentioned above. We apologize for this oversight and would like to express our gratitude for the opportunity to submit a further revision of our manuscript.

---

## [Editor Report · Decision Letter 2]

7 Mar 2024

Reciprocity in Ambiguous Situations: Default Psychological Strategies Underlying Ambiguity-Resolution in Moral Decision-Making

PONE-D-23-24513R2

Dear Dr. Galvan,

We’re pleased to inform you that your manuscript has been judged scientifically suitable for publication and will be formally accepted for publication once it meets all outstanding technical requirements.

Kind regards,

Inon Zuckerman

Academic Editor

PLOS ONE
---

## [Editor Report · Acceptance letter]

27 Mar 2024

PONE-D-23-24513R2 

PLOS ONE

Dear Dr. Galvan, 

I'm pleased to inform you that your manuscript has been deemed suitable for publication in PLOS ONE. Congratulations! Your manuscript is now being handed over to our production team.

Kind regards, 

on behalf of

Dr. Inon Zuckerman 

Academic Editor

PLOS ONE